# A Systematic Review and a Meta-Analysis of the Yellow Fever Vaccine in the Elderly Population

**DOI:** 10.3390/vaccines10050711

**Published:** 2022-04-30

**Authors:** Ariane de Jesus Lopes de Abreu, João Roberto Cavalcante, Letícia Wigg de Araújo Lagos, Rosângela Caetano, José Ueleres Braga

**Affiliations:** 1Instituto Nacional de Cardiologia, Rio de Janeiro 22240-006, Brazil; letwigg@terra.com.br; 2Instituto de Medicina Social da Universidade do Estado do Rio de Janeiro, Rio de Janeiro 20550-013, Brazil; joao.rcs@hotmail.com (J.R.C.); caetano.r@gmail.com (R.C.); ueleres@gmail.com (J.U.B.); 3Escola Nacional de Saúde Publica Sergio Arouca-FIOCRUZ, Rio de Janeiro 21041-21, Brazil

**Keywords:** yellow fever vaccine, adverse events, systematic review, aged

## Abstract

We conducted a systematic review and a meta-analysis to assess the risk of serious adverse events in the elderly after yellow fever vaccination compared to the non-elderly population. We searched multiple databases and grey literature, and we selected research without language and publication date restrictions. Studies were analyzed in a descriptive way and meta-analyzed and expressed in terms of prevalence ratio and risk ratio with a 95% confidence interval, depending on the degree of heterogeneity found. A total of 18 studies were included and 11 were meta-analyzed. The results obtained through the meta-analysis showed a risk of serious adverse events after yellow fever vaccination three times higher for the elderly when compared to the non-elderly population and five times higher for persons > 70 years. In relation to adverse event types, viscerotropic disease associated with the yellow fever vaccine had a risk that was six times higher when compared to the population < 60 years. The evidence found supports that the vaccine indication in individuals > 60 years of age should be based on a careful analysis of individual benefit-risk assessments. The results found suggest a higher risk of events for individuals > 70 years, especially for viscerotropic and neurotropic disease associated with YFV contraindicating the use of the YFV in this age group.

## 1. Introduction

The yellow fever vaccine (YFV) is the most important means of preventing the disease. It is an immunobiological product considered safe and effective, requiring only one dose to ensure protection against the disease [1]. However, some situations may present a higher risk of adverse events, requiring an individualized assessment of the risk-benefit to recommend the vaccine, as is the case with people over 60 years of age, due to the natural process of immunosenescence [2,3,4,5]. Another important issue is that elderly people with comorbidities are more susceptible to infectious diseases. Once infected, there is a risk of suffering an exacerbation of clinical symptoms, which can cause other complications and death [6].

In the last decades, emergency/re-emergence processes of the yellow fever virus have had an important impact on human and animal populations, represented by extensive outbreaks in humans and epizootics in non-human primates, such as that which affected Angola and the Democratic Republic of Congo in 2015 and 2016 [7,8,9,10]. From 2017 to December 2020, other outbreaks were reported in Nigeria, Senegal, Suriname, Guinea, French Guiana, Democratic Republic of Congo, Venezuela, Male, Ethiopia, Togo, Gabon, Liberia, Uganda, and Brazil [11]. The recent high numbers of yellow fever cases were attributed to low vaccination coverage—lower than that needed to prevent outbreaks. Additionally, the increase in global travel and population movements pose an increased risk of introduction into large urban areas in tropical and subtropical areas that are infested with mosquitoes competent to transmit YF [8,10].

According to the World Health Organization, by 2020 the number of people over 60 will surpass that of children under 5 years for the first time in history [12]. With the increase in life expectancy of the world population and the demographic transition process resulting in an aging population, it is necessary to discuss prevention and health promotion measures for the elderly [13]. In the current scenario, this includes forms of prevention against yellow fever, such as vaccination.

YFV is recommended for people over 60 years of age through a risk-benefit assessment. Due to the phenomenon of immunosenescence, the risk of adverse events occurring in the elderly is greater than that of young adults [14,15]. The most serious adverse event (SAE) linked with YFV in the elderly is viscerotropic disease, which is the spread of the vaccine virus to several organs, with shock, pleural and abdominal effusion, and multiple organ failure [15,16].

There are systematic reviews that address the analysis of YFV safety in the elderly, however, these included only in articles published until 2012, before the latest major outbreaks of the disease in the world and the increase in vaccine recommendation areas [15,17,18,19]. These reviews concluded that there is a greater risk among the elderly, with a significant increase in elderly travelers for YFV-related adverse events compared to other age groups. However, they argue that the existing evidence on this topic is limited and that more research on the subject is needed. Therefore, we conducted a systematic review to assess the safety of YFV in the elderly.

## 2. Materials and Methods

This systematic review was reported according to the harms checklist from the Preferred Reporting Items for Systematic Reviews and Meta-Analyses (PRISMA harms) guidelines [20] (Appendix A). The review guiding question was ‘what is the risk of SAE in adults over the age of 60 years after YFV vaccination and when compared with the population under 60 years?’. We included studies published until December 2021, with no geographic restriction and published in English, Spanish, French, or Portuguese. The SAE outcomes considered were anaphylaxis, neurological manifestations associated with YFV, vaccine-associated viscerotropic disease, and death. This review was registered in the International Prospective Register of Systematic Reviews (PROSPERO) with number CRD42020160430.

### 2.1. Information Sources and Search Strategy

We searched PubMed/Medline, Embase, Web of Science, Scopus, Lilacs, the Database of Abstracts of Review of Effects, and Toxiline as well as sources of grey literature (Open grey, Grey literature report, the CDC Morbidity and Mortality Weekly Report, the Australian Adverse Drug Reactions Bulletin, the European Database of Suspected Adverse Drug Reaction Reports, the Institutional Repository for Information Sharing of the World Health Organization, the annals of the São Paulo Congress of Infectiology, and ClinicalTrials.gov, accessed on 7 November 2021). Those databases were chosen based on the repositories consulted in the previous systematic reviews on the topic and if they were considered reference repositories related to adverse events publication. We used a combination of search terms related to the population, YFV, and outcomes—considering each database’s specific descriptors and free text to increase the search strategy sensitivity as provided in Appendix A. The search was performed in September 2020 and updated in December 2021. Cross references that fulfil this review’s inclusion and exclusion criteria were also included.

### 2.2. Information Sources and Search Strategy

We included studies with original data on SAE from YFV in adults aged more than 60 years, which had a population denominator (as the total number of doses applied for the population). Study designs considered for inclusion were clinical trials, cohort, case-control, and case series. All different lineages of YFV were considered for inclusion. Studies were excluded if they referred to multiple publications. In that case, only the publication with the most current data was considered.

Two independent reviewers (A.A. and L.W.) critically appraised each paper and discussed discrepancies in consensus. We screened titles and abstracts according to specified inclusion and exclusion criteria and then selected the full texts.

Data was extracted independently by two reviewers using an electronic form created in Epidata version 4.6. A pilot was conducted with 10% of the studies included in the review.

To deal with potential data gaps, authors from the included studies were contacted in up to three attempts by email to obtain unreported or unclear data. If no reply was obtained the study was excluded from the review.

### 2.3. Information Sources and Search Strategy

The quality of included studies was assessed in accordance with the methodology recommended in the Cochrane Handbook for Systematic Reviews of Interventions using both the Revised Cochrane risk-of-bias tool for randomized trials (RoB 2.0) and the Risk of Bias in Non-randomized Studies-of Interventions (ROBINS-I) tools [21,22]. Two independent reviewers (A.A. and J.C.) critically appraised each study and discussed discrepancies in consensus.

The GRADE methodology was used to assess the quality of the evidence of the review [23], using the GRADEpro tool.

### 2.4. Information Sources and Search Strategy

Data extracted from the studies were characterized depending on the homogeneity of the study planning methods and outcomes. An evaluation of heterogeneity was carried out using the I^2^ test. The level of heterogeneity was defined as low (I^2^ = 0% to 25%), moderate (I^2^ ≥ 25% to 75%), and high (I^2^ ≥ 75%) [24]. The chi-square test, with a significance level of *p* = 0.05, was also performed. If heterogeneity were above moderate, a narrative analysis summarizing the findings of the studies was conducted using a random effects model [24,25].

A meta-analysis was conducted to assess the relative frequencies of SAE in persons > 60 years when compared to those aged under 60 years, using the prevalence rate and 95% confidence interval. Additionally, a meta-analysis of risk factors was conducted for the included studies that presented measures of association, the groups with SAE outcomes were compared with the control group through the risk ratio and the 95% confidence interval.

Sensitivity analyses were performed to examine potential explanations of the heterogeneity found between studies by factors related to clinical and methodological characteristics of the studies, such as analysis after removal of outliers, subgroup analysis, and meta-regression. To examine publication bias, the Egger test and the Begg test were performed, as well as an analysis of the funnel graph. The program R version 4.0.2 was used for statistical analysis of the data.

## 3. Results

We identified 1418 published studies from database searches. No articles were found in the DARE, Open Gray, Gray literature report, Australian Adverse Drug Reactions Bulletin, and FAERS databases.

After removing duplicated publications, the first step of selecting articles by reading the titles and abstracts identified 47 papers for full text reading. In this first stage, 14 publications per language were excluded, 6 in Russian, 3 in Arabic and 5 in Chinese.

Eighteen studies were identified for analysis [4,18,26,27,28,29,30,31,32,33,34,35,36,37,38,39,40,41] of which eleven presented the necessary calculation information to perform the meta-analysis (Figure 1).

### 3.1. Study Characteristics

The studies included were published between 2001 and 2020, most with cross sectional design (61.1%) and with a period of data collection greater than 3 years (50.0%). Only three studies were published after the latest yellow fever outbreaks since 2015. Only three studies (16.7%) presented information for a specific age group of the elderly population and 9 (55.6%) did not present information for comorbidities (Table 1).

Regarding the country of conduct of the selected studies, four were multi-country studies [28,29,30,32]. The most frequent countries analyzed by the studies included in the review were the United States (33.3%) and Brazil (18.5%).

In total, 322,759,087 doses of YFV were administered in the studies evaluated. Appendix A shows the characteristics of the interventions assessed in the studies included in the review. The most frequent vaccine strain among the studies was 17D (38.9%). Most studies did not have information on the type of dose administered (88.9%) and the type of vaccination performed (61.1%). In total, eight studies (44.4%) evaluated co-administration with other vaccines, most of them with Hepatitis A (24.2%) and typhoid fever (18.2%), generally related to traveler vaccination.

Of the studies included in the review, eight did not have information to enable the combined analysis of the data and, therefore, were excluded from the meta-analysis [26,28,30,31,38,39,41]. The total number of serious adverse events present in those seven studies is summarized in Appendix A. No cases of anaphylaxis were reported in the studies presented.

The samples of the studies included in this review presented data by the number of participants or the distributed/administered doses. Thus, the prevalence was adopted as a frequency measure for the meta-analysis of the 10 studies [4,18,29,32,33,34,35,36,37,40] evaluated (Appendix A). The meta-analysis showed a prevalence of 32 cases for every 1,000,000 vaccinated (95% CI 0.006–0.171). However, given the high risk of heterogeneity between the studies presented by the value of I^2^ = 98% and the chi-square (*p* < 0.01) as well as the visual highlight presented in the forest plot of one study as an outlier [36], a new analysis was performed with the removal of this study (Figure 2). Despite a small reduction in heterogeneity (I^2^ = 96% and *p* < 0.01), there was an increase in the value found for prevalence, with a value of 14 cases for every 1,000,000 vaccinated (CI 95% 0.003–0.068).

Of the ten studies evaluated, seven [4,18,32,33,34,35,36] had data referring to a comparison group (age group < 60 years), and they were evaluated through meta-analysis for risk factors for SAE (Figure 3). The result of the analysis for heterogeneity (I^2^ = 60%, *p* = 0.02) indicates that there are important differences between the results of the studies for this outcome. They identified that the risk was 2.51 times higher for the age group of people ≥ 60 years old (95% CI 1.71–3.69) when compared to the population < 60 years old.

To explore possible clinical and methodological factors that could explain the degree of heterogeneity presented in the previous meta-analysis, a subgroup analysis was performed, considering factors related to population, intervention, and study methodology.

A subgroup analysis was conducted considering the different age groups of the elderly population to explore the high heterogeneity found in the previous analysis and a potential “dose-response” relationship between the age groups (Figure 4A). The analysis showed greater homogeneity between studies for the groups < 70 years (I^2^ = 30%, *p* = 0.20) and ≥70 years (I^2^ = 23%, *p* = 0.27). Overall, the analysis reveals that the risk of SAE increases with age. The risk of SAE found was 2.32 times higher for <70 years old (95% CI 1.44–3.72) and 4.84 times higher for the age group ≥ 70 years old (95% CI 2.82–8.31) when compared to the population < 60 years old.

The type of SAE reported was also evaluated in a subgroup analysis (Figure 4B). There was a reduction in heterogeneity (I^2^ = 44%; *p* = 0.17) for studies that reported SAE related to viscerotropic disease associated with YFV. For this group, the risk was 6.59 times higher (95% CI 2.14–20.28) for elderlies when compared to the group aged <60 years. For neurological manifestations associated with YFV, although heterogeneity remained high (I^2^ = 79%; *p* < 0.01), the risk was 4.45 times higher (95% CI 1.14–17.38).

Moreover, a subgroup analysis for the study design type was performed (Figure 4C). The degree of heterogeneity was high for the longitudinal study group (I^2^ = 89%; *p* < 0.01), which can be explained by their distinct study designs (cohort [18] and experimental [36]). Other subgroup analysis was also performed for comorbidity presence, vaccine lineage used, and coadministration. However, the data available in the included studies were not enough to perform the analysis with details that could find a clinical or methodological explanation for the heterogenicity found.

A meta-regression was performed to examine the individual contributions of the studies in the heterogeneity found and the potential associations with age, SAE type, and study design (Appendix A). Age was the characteristic that was associated with the heterogeneity initially found in the risk factor meta-analysis. The age groups < 70 years (*p* = 0.0250) and the elderly without age specification (*p* = 0.0089) showed a greater association with the heterogeneity found when compared to the age group above 70 years. No association was found between heterogeneity and the characteristics assessed for study design and type of SAE.

As for the publication bias analysis, for the prevalence meta-analysis both the asymmetry shown in the funnel graph and the result of the Begg test (*p* = 0.2637) suggest publication bias, indicating a greater uptake of studies that showed a greater effect related to the risk of SAE. For the risk factor meta-analysis, the symmetry found in the funnel graph and the result found by both tests (*p* = 0.5961) do not suggest a publication bias.

### 3.2. Risk of Bias within Studies and across Studies

Considering the quality of the studies included in the review, no study presented a low risk of bias for the domains evaluated with the ROBINS-I. Most studies did not provide enough information to judge the different types of bias, mainly confounding (87.5%), not receiving the assigned intervention (87.5%), and measuring the intervention (81.2%). The domain with the highest number of studies with a critical bias level was selection (56.2%) (Appendix A). Only one study included in the review had a randomized clinical trial as a study design; it was assessed using the RoB 2.0 tool and it presented a moderate risk of bias for most domains (Appendix A). The quality of the evidence analyzed through the GRADE system from the review data suggests low and moderate confidence in the estimated effect from the studies included in the review.

## 4. Discussion

In general, the findings of this literature review showed a higher risk of SAE for the population over 60 years, and the elderly population had a risk three times higher of developing SAE when compared to the non-elderly population.

Furthermore, the results obtained through the meta-analysis showed that this risk is almost five times higher for the age group over 70 years old, when compared to those aged 60 to 69 years. These findings are in line with previous research on this topic [18,19,36], although none had meta-analyzed data. Thomas et al. (2012) reports that only five passive surveillance databases identified a small number of cases of yellow fever vaccine-associated viscerotropic disease, yellow fever vaccine-associated neurotropic disease, and anaphylaxis in persons ≥ 60 years, which is aligned with the result found in the meta-analysis of prevalence ratio. In accordance with our results for the comparison of SAE after yellow fever vaccination between the elderly and the non-elderly populations, Khromava et al. (2005) found that the reporting rates of serious adverse events were significantly higher among vaccinees aged ≥60 years than among those 19–29 years of age (reporting rate ratio = 5.9, 95% CI 1.6–22.2) for both viscerotropic disease and neurological events after yellow fever vaccination. A similar result was found by Monath et al. (2005) in which the incidence of significant neurologic and multisystem AEs reporting ratio was 3-fold higher for the 65–74-year-old age group (Report Risk Ratio 2.82; 95% CI 0.81, 9.81).

Among the types of SAE analyzed in the review, the greatest risk was for viscerotropic disease associated with YFV up to six times higher when compared to the population under 60 years. Other studies carried out for this target population found a similar tendency of a higher risk for both viscerotropic diseases associated with YFV and neurotropic disease associated with YFV compared to other types of SAE [15,18,38,42].

Concerning the presence of heterogeneity found, subgroup analysis and meta-regression showed that age is associated with the degree of heterogeneity found in the meta-analysis of risk factors. This may be related to some of the characteristics of the immunosenescence process, in addition to other possible individual characteristics of the elderly <70 years old present in the studies when compared to those aged ≥70 years old [2,3,4,5].

As for the methodological quality of the studies included in the review, this ranged from moderate to critical depending on the assessment tool used. The most frequently included study designs were non-experimental, such as those that were transversal. This characteristic can also be associated with the high degree of heterogeneity found in the meta-analysis, since the highest risks of bias were related to the selection of participants and losses.

The same was reflected in the general analysis of the quality of the review′s evidence that suggests low and moderate confidence in the estimated effect from the included studies. Despite this, it was observed that the evidence evaluated in this study has a strong recommendation against the use of YFV in the target population of the review because it presents a higher risk of SAE, especially in the age group above 70 years.

Our study has some limitations. The studies included in this review may have different bias affecting the denominators used, among them the differences between the study designs, the selection of the participants, the loss of information, and the participants, which directly influences the confidence of the conclusions generated in the results found. The same also occurred for the numerators used since the classification of some types of SAE varied between studies.

Another limitation is related to the difficulty of investigating SAE for this specific population. This is because the events found may be associated with other pre-existing factors for these individuals, such as comorbidities, which can make it challenging to prove causality between the application of the vaccine and the involvement of the adverse events itself [14,19,43]. It is also important to highlight that misclassification of cases due to flavivirus cross-reactivity could occur during the investigation of SAE, as tests for other flavivirus were not always done, leading to cases that could have been incorrectly classified as related to YFV [35].

The yellow fever vaccine research program has some inherent limitations which may be reflected in the findings of this review. Although we developed a sensitive search strategy and included several database sources, few publications were found on this topic for this specific population. Furthermore, only three studies included were published after the latest outbreaks of the disease, since 2015.

This fact can be related to two main factors. First, YFV is not routinely recommended for the target population of this review, but only under medical judgment [1,2]; secondly, YFV is considered an effective old technology and widely used, which has generated over the years a confidence in the safety of the vaccine [44].

Thus, the evidence found in this review supports the indication of YFV in individuals over 60 years of age must be based on a careful analysis of an individual benefit-risk assessment. However, it is necessary to highlight that the results found suggest a higher risk of SAE for individuals over 70 years, especially for viscerotropic and neurotropic disease associated with YFV contraindicating the use of the YFV in this age group.

It is noteworthy that the individual benefit-risk analysis must be carried out with a particular perspective in regions where the epidemiological scenario reinforces a high risk of contracting the disease, such as in endemic regions, where the population is naturally exposed for a longer period. This question should be taken as a basis for users, health professionals, and other decision makers on the use of this technology. Advances in the development of new yellow fever vaccines could be a strategy to avoid serious adverse events. One approach could be inactivated vaccines currently under early-phase clinical studies [45].

Yellow fever is a serious public health concern and without the YFV the risk of an increase in outbreaks and large-scale epidemics would be high, mostly related to the extensive spread of disease vectors and due to an ever-increasing movement of people around the globe. Even though YFV adverse events need to be considered, the vaccine is the best way to avoid the disease, and it should be used in endemic areas and in populations that could potentially be exposed, considering a careful risk-benefit analysis. More research to increase knowledge about the potential risk of SAE in specific contexts for the population studied and to generate new vaccine technologies for the disease, such as inactivated vaccines against YF already underway, is needed to guide the best recommendation for this population.

## Figures and Tables

**Figure 1 vaccines-10-00711-f001:**
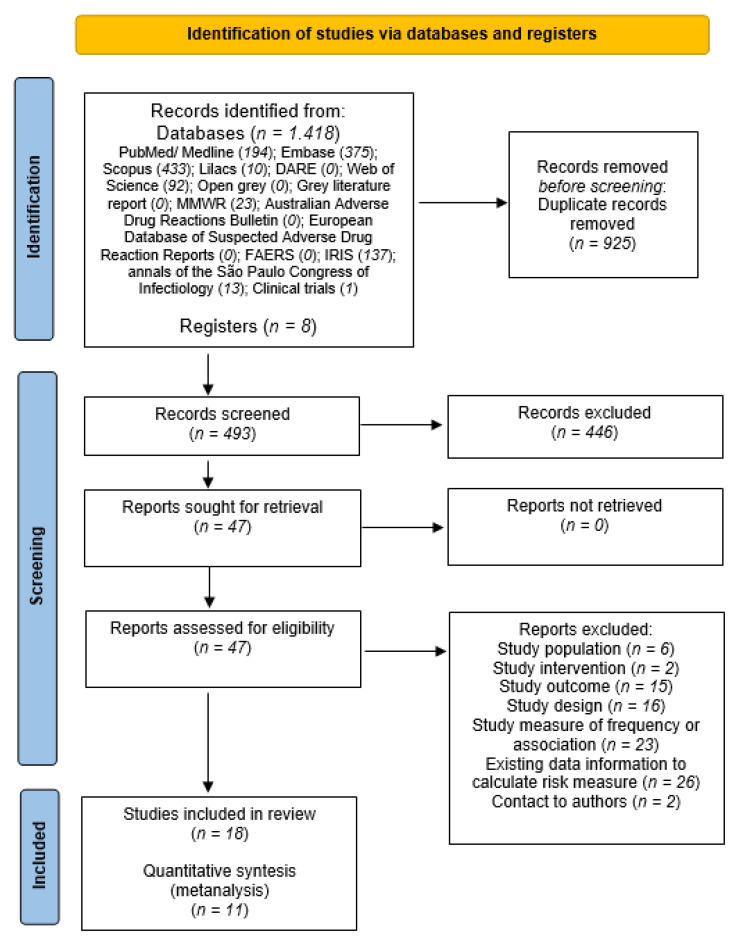
Process of study selection of systematic review and meta-analyses.

**Figure 2 vaccines-10-00711-f002:**
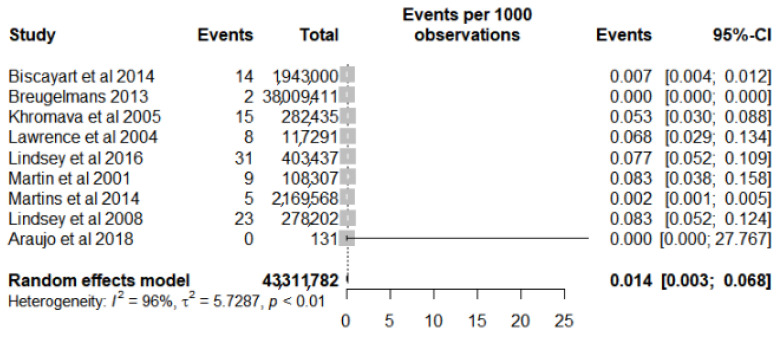
Forest plot of prevalence meta-analysis for serious adverse events after yellow fever vaccine use in the elderly after outlier study removal.

**Figure 3 vaccines-10-00711-f003:**
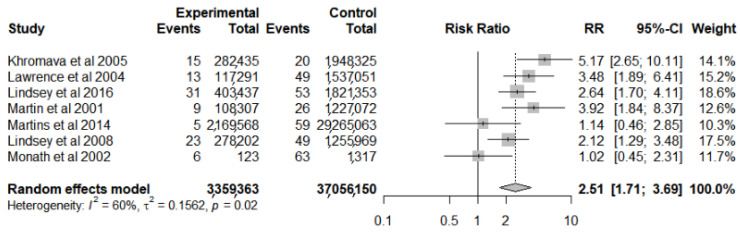
Forest plot of risk factor meta-analysis for serious adverse events after yellow fever vaccine use in the elderly.

**Figure 4 vaccines-10-00711-f004:**
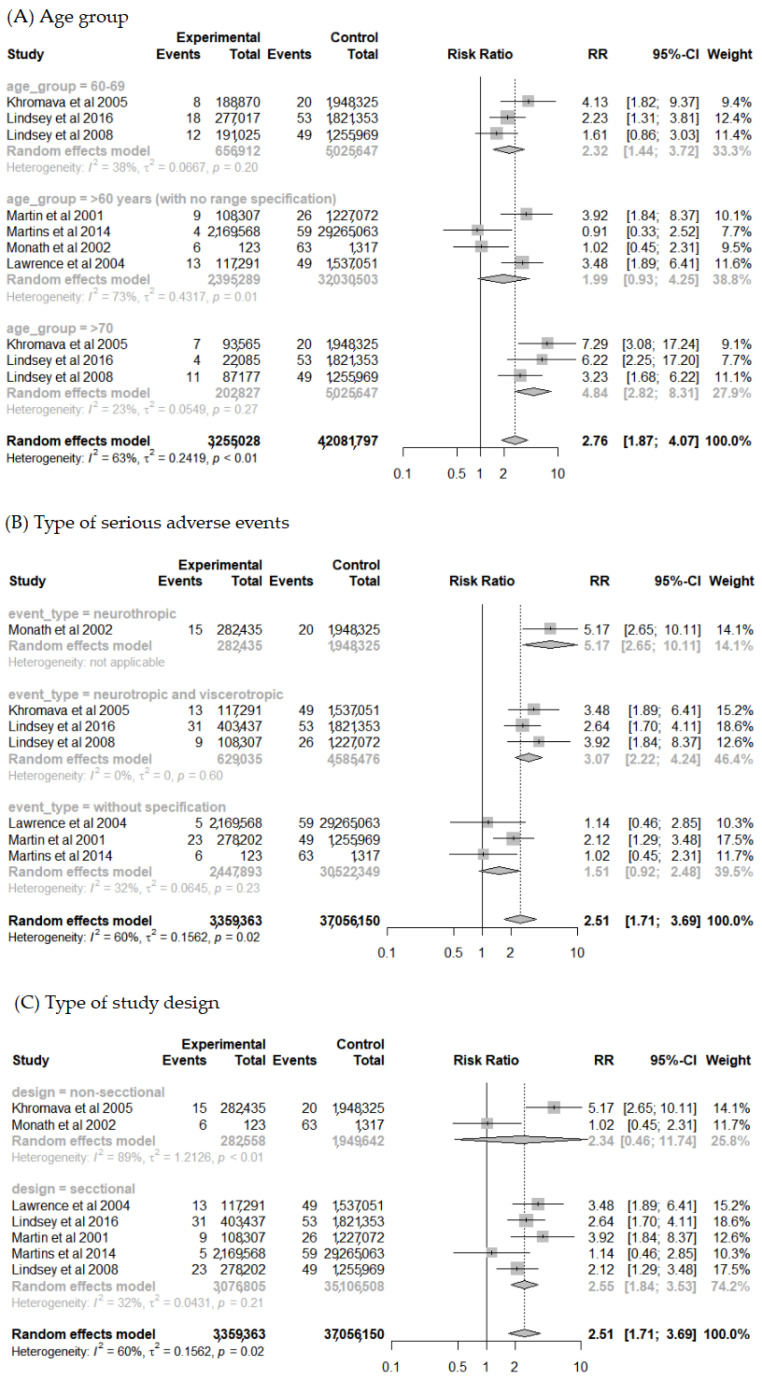
Forest plot of risk factor meta-analysis for adverse events after yellow fever vaccine use in participant subgroups by: (**A**–**C**).

**Table 1 vaccines-10-00711-t001:** Characteristics of the studies included in the review.

Author	Journal	Data Collection Period	Study Design	Follow up Time	Total Sample Size	Elderly Age Group Assessed	Total Number of Elderlies Assessed	Comorbidities Presented in the Studies	Included in the Meta-Analysis
CDC, 2002 [26]	Morbidity and Mortality Weekly Report	>1 to 3 years	Case series	>1 to 3 years	6	70 to 79 years	2	History of cardiovascular disease	No
Azevedo et al. 2011 [27]	Transplant Infectious Disease Journal	Missing information	Cross sectional	Missing information	19	60 to 69 years	2	History of cardiovascular disease	No
Bae et al. 2008 [28]	The Journal of Infectious Diseases	>3 years	Case series	<6 months	6	60 to 69 years	3	Missing information	No
Biscayart et al. 2014 [40]	Vaccine	6 months to 1 year	Cross sectional	6 months to 1 year	165	>60 years	7	History of cardiovascular disease and allergies	Yes
Breugelmans, 2013 [29]	Vaccine	>3 years	Cross sectional	>3 years	3116	>60 years	2	Missing information	Yes
Cottin et al. 2013 [30]	Expert review of vaccines	>3 years	Cross sectional	>3 years	1460	>60 years	Missing information *	History of cardiovascular disease and chronic kidney disease	No
Mota et al. 2009 [31]	Revista da Sociedade Brasileira de Medicina Tropical	Missing information	Cross sectional	Missing information	70	>60 years	3	History of immunosuppressive disease	No
Khromava et al. 2005 [18]	Vaccine	>3 years	Cohort	>3 years	722	>60 years	58	Missing information	Yes
Lawrence et al. 2004 [32]	Communicable Diseases Intelligence Quarterly Report	>3 years	Cross sectional	<6 years	42	>60 years	Missing information *	Missing information	Yes
Lindsey et al. 2016 [33]	Journal of Travel Medicine	>3 years	Cross sectional	>3 years	938	>60 years	Missing information *	Missing information	Yes
Martin et al. 2001 [38]	Emerging infectious diseases	>3 years	Cross sectional	>3 years	5125	>60 years	285	Missing information	Yes
Martins et al. 2014 [35]	Vaccine	>3 years	Cross sectional	>3 years	67	>60 years	Missing information *	History of immunosuppressive disease	Yes
Lindsey et al. 2008 [4]	Vaccine	>3 years	Cross sectional	>3 years	660	>60 years	97	Missing information	Yes
Monath et al. 2002 [36]	The American journal of tropical medicine and hygiene	<6 months	Randomized clinical trial	<6 months	1440	>60 years	123	Missing information	Yes
Araujo et al. 2018 [37]	The Brazilian Journal of Infectious Diseases	6 months to 1 year	Cohort	6 months to 1 year	131	>60 years	131	History of immunosuppressive and cardiovascular diseases and diabetes	Yes
Martin et al. 2001 [38]	The Lancet	6 months to 1 year	Case series	<6 months	4	>60 years	4	History of cardiovascular disease and chronic kidney disease	No
McMahon et al. 2006 [39]	Vaccine	6 months to 1 year	Cross sectional	6 months to 1 year	15	>60 years	6	Missing information	No
Lucena et al. 2020 [41]	Epidemiologia e Serviços de Saúde	>1 to 3 years	Case control	>1 to 3 years	NA	>60 years	NA	Missing information	No

* The study assessed the elderly but did not present the total sample size for this population in the text; NA: not applicable (population not available; only adverse events number were presented).

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
