# Peer review of "A Systematic Review and a Meta-Analysis of the Yellow Fever Vaccine in the Elderly Population"

_vaccines, 2022, doi:10.3390/vaccines10050711_

Round 1
Reviewer 1 Report
Estimated Authors of this very interesting Systematic Review and Meta-analysis on YFV in elderly,
first of all thank you for the considerable efforts you've performed in order to collect and summarize available evidence. According to your analyses, adverse effects after YFV in elderly would occur in around 1.4 cases per 100,000 persons, with a risk that is nearly three times that occurring in younger individuals. Even though these data are substantially consistent with older reviews, I think that your paper may share appropriate understanding on a major issue with a very important vaccine, not only for individuals dwelling in high risk areas, but also for international travelers.
From my point of view, I've only a couple of suggestions to improve the relatively high quality of your research:
1) according to table 2, a substantial share of the paper included in the review lack of several informations including the vaccine lineage. You should clarify whether the paper included in the meta-analyses were among those that lacked of such information or not. Moreover, please adjust Table 1 by removing the column on the journal that hosted the article, and replacing with the information about the inclusion or not of the paper in the meta-analysis. Moreover, I warmly suggest to segregate the case series either in a separeted summary or in a distinctive section of table 1.
2) Data included in supplementary materials are often of valuable relevance, and could be rather included in the main text as annexes. More precisely, I'm speaking about Figure S3, and Figure S4.
Finally, please avoid the abbreviation "AE" as it is (I think) not necessary, while your could retain SAE.
Author Response
Dear reviewer,
We strongly appreciate all of your suggestions and tried to implement them in the best way possible as stated below:
1) In relation to the articles with lack of data/ higher proportion of missing information from table 2, we stated that the articles were included in the meta-analysis and that we performed a subgroup analysis considering those characteristics to verify its impact on the heterogenicity found in the results. This is stated in lines 231-234: "Other subgroup analysis was also performed for comorbidity presence, vaccine lineage used and coadministration. However, the data available in the included studies were not enough to perform the analysis with details that could find a clinical or methodological explanation for the heterogenicity found."
As for your comment related to table 1, we have added a new column demonstrating which articles were included or not in the meta-analysis. We chose to maintain the journal column to allow the readers to compare it better with the previous systematic reviews on this topic that also provided this type of results. We also maintained the cases series in table one, since its a general table to provide the overall characteristics of the articles included. We believe this will facilitate for the readers to have an overall picture. Further details related to results from case series studies are provided in table S4.
2) In relation to figures S3 and S4, we really appreciate your suggestion and we aggregated them and figure S2 into new figure 4. We believe this will highlight better all the subgroup analysis results.
3) Thank you for the suggestion related to the abbreviation of AE in the text. Those are now removed and only SAE was kept.
Reviewer 2 Report
The aim of the article “Systematic review and meta-analysis of yellow fever vaccine in elderly population” was to assess the risk of serious adverse events (SAE) in the elderly after yellow fever vaccination compared to the non-elderly population (< 60 years old). The theme is relevant and the results interesting, and according to previous reports. The article is well written and, in my opinion, should be published. Additionally, authors highlight the limitations of the study in the discussion section, which is very valuable. A minor details and suggestions are indicated below:
- In line 78, authors indicate which databases were consulted. They included Medline, Embase, Web of Science, Scopus, Lilacs, Database of Abstracts of Review of Effects and Toxiline, as well as sources of grey literature. However, important databases such as Science Direct or Pubmed were not consulted. Please, indicate the criteria for choosing the databases.
- In “Figure 1. Process of study selection of systematic review and meta-analyses”, the diagram incorporates the total records identified from databases and records removed, however, the record identified in each database was not indicated.
- Line 140 authors indicate that 14 publications were excluded per language, 6 in Russian, 3 in Arabic and 5 in Chinese. However, previously, in line 73, authors indicate that “We included studies published until December 2021, with no language or geographic restriction”.
- I consider that this figure and result (S2) is relevant and it should be incorporated as a result and not as a supplementary material.
Author Response
Dear reviewer,
We are very thankful for your suggestions. We tried to implement them in the best way possible as stated below:
1) Figure 1 is now modified to include results found per database. We also modified the text as suggested clarifying the database selection process for this review. It is presented at lines 80-88: "We searched Pubmed/Medline, Embase, Web of Science, Scopus, Lilacs, Database of Abstracts of Review of Effects and Toxiline, as well as sources of grey literature (Open grey, Grey literature report, CDC Morbidity and Mortality Weekly Report, Australian Adverse Drug Reactions Bulletin, European Database of Suspected Adverse Drug Reaction Reports, Institutional Repository for Information Sharing of the World Health Organization, annals of the São Paulo Congress of Infectiology and ClinicalTrials.gov). Those databases were chosen based on the repositories consulted in the previous systematic reviews on the topic and if they were considered reference repositories related to adverse events publication."
2) In relation to your comment regarding line 140 and eligibility criteria we modified in lines 73-74 to include the limit related to language.
3) In relation to figure S2, we modified it and aggregated figures S2, S3 and S4 into new figure 4 in the main manuscript. Previous table 2 is now presented as Table S3. We also agree this will better highlight the results found in our research.